# Risk of Dementia after Exposure to Contrast Media: A Nationwide, Population-Based Cohort Study

**DOI:** 10.3390/biomedicines10082015

**Published:** 2022-08-19

**Authors:** Tung-Min Yu, Ya-Wen Chuang, Shih-Ting Huang, Jin-An Huang, Cheng-Hsu Chen, Mu-Chi Chung, Chun-Yi Wu, Pi-Yi Chang, Chih-Cheng Hsu, Ming-Ju Wu

**Affiliations:** 1Division of Nephrology, Department of Internal Medicine, Taichung Veterans General Hospital, 1650 Taiwan Boulevard Sect. 4, Taichung 40705, Taiwan; 2Graduate Institute of Biomedical Sciences, College of Medicine, China Medical University, Taichung 404333, Taiwan; 3School of Medicine, College of Medicine, China Medical University, Taichung 404333, Taiwan; 4Department of Post-Baccalaureate Medicine, College of Medicine, National Chung Hsing University, Taichung 40227, Taiwan; 5Department of Neurological Institute, Taichung Veterans General Hospital, Taichung 40705, Taiwan; 6Department of Health Business Administration, Hungkuang University, Taichung 43302, Taiwan; 7Department of Radiology, Taichung Veterans General Hospital, Taichung 40705, Taiwan; 8Institute of Population Health Sciences, National Health Research Institutes, Zhunan, Miaoli 35053, Taiwan; 9Department of Family Medicine, Min-Sheng General Hospital, Taoyuan 33044, Taiwan; 10Rong Hsing Research Center for Translational Medicine, National Chung Hsing University, Taichung 40227, Taiwan; 11Ph.D. Program in Translational Medicine, National Chung Hsing University, Taichung 40227, Taiwan; 12School of Medicine, Chung Shan Medical University, Taichung 40201, Taiwan

**Keywords:** dementia, contrast medium, Alzheimer’s disease, vascular dementia

## Abstract

Contrast-medium-associated kidney injury is caused by the infusion of contrast media. Small vessel disease is significantly associated with various diseases, including simultaneous conditions of the kidney and brain, which are highly vulnerable to similar vascular damage and microvascular pathologies. Data to investigate the adverse effect of contrast media on the brain remain extremely lacking. In this study, 11,332,616 NHI enrollees were selected and divided into two groups, exposed and not exposed to a contrast medium during the observation period, from which 1,461,684 pairs were selected for analyses through matching in terms of age, sex, comorbidities, and frequency of outpatient visits during the previous year. In total, 1,461,684 patients exposed to a contrast medium and 1,461,684 controls not exposed to one were enrolled. In multivariable Cox proportional hazard models, patients exposed to a contrast medium had an overall 2.09-fold higher risk of dementia. In multivariable-stratified analyses, the risk of Alzheimer’s disease was remarkably high in younger patients without any underlying comorbidity. This study is the first to discover that exposure to contrast media is significantly associated with the risk of dementia. A four-fold increased risk of vascular dementia was observed after exposure to a contrast medium. Further studies on the influence of exposure to contrast media on the brain are warranted.

## 1. Introduction

Contrast medium (CM)-induced renal failure is the third most common cause of hospitalization and mortality in patients with acute kidney injury (AKI), accounting for 11% of all cases [1]. Contrast agents can be either directly toxic to renal tubular epithelial cells, resulting in renal dysfunction and both necrosis and apoptosis, or indirectly lead to potent vasoconstriction within intrarenal small vessels, resulting in low glomerular blood flow and oxygen delivery to the nephron. Moreover, contrast agents may increase the viscosity and osmolarity of the blood, which can destroy the plasticity of erythrocytes, consequently increasing the risk of microvascular thrombosis [2].

An emerging line of evidence suggests that CMs may likely result in injury to neuronal cells. Patients occasionally experience neurological symptoms such as nausea, headache, and painful sensation after the intravascular infusion of CMs, which is associated with their hyperosmolar nature, especially at the neuronal level [3].

An accumulating body of evidence suggests that both the kidney and brain are vulnerable to similar vascular damage and microvascular pathologies. They can be mutually affected by small vessel disease (SVD) in a similar manner; SVD adversely affects the arteriole, venules, and capillaries, leading to ischemia and hemorrhagic lesions [4,5].

Dementia is a global health concern and is predicted to affect approximately 81.1 million individuals worldwide by 2040 [6]. Alzheimer’s disease (AD) is the most common type of dementia, accounting for 60% of all dementia cases, with a 3.4% mean prevalence, and vascular dementia (VD) is the second most common type of dementia, particularly in individuals aged over 65 years [7,8].

Multiple factors are likely associated with the development of AD as well as other types of dementia, such as age, sex, education level, and the condition of having an infectious disease, malnutrition, traumatic brain disease, or depressive illness [7,9]. Moreover, some atherosclerosis factors are likely associated with an increased risk of AD as well as VD, such as hypertension, dyslipidemia, diabetes mellitus, obesity, and subclinical atherosclerosis [7]. Vascular diseases, particularly small vessel disease (SVD), and neurodegeneration are intricately interrelated [10]. For example, in cases with VD, a growing body of evidence suggests that cerebral SVD may be the most common contributor to dementia, which is characterized by arteriosclerosis and subcortical microinfarcts, the most common substrate of cognitive impairment [11,12].

Data to investigate the risk of dementia and the history of exposure to CMs are very limited. Thus, the aim of this study was to determine the risk of dementia after exposure to CMs in a nationwide population-based cohort.

## 2. Methods

### 2.1. Data Sources

More than 450 hospitals and 10,000 clinics have been providing health care services reimbursed by the National Health Insurance (NHI) program in Taiwan since 1995. The NHI Administration established the NHI Research Database (NHIRD), which contains information regarding the health care services received by all NHI enrollees (>99% of 23 million Taiwanese residents), including disease diagnoses, laboratory tests, drug prescriptions, operation codes, medical procedures, and medical costs in all ambulatory visits and hospitalizations. The NHIRD, which was used as the data source in this study, is the most comprehensive database available in Taiwan for health care research. Disease diagnoses and procedures in the NHIRD were coded according to the International Classification of Diseases, 9th Revision, Clinical Modification (ICD-9-CM). The NHIRD was also linked to the National Death Registry. Personal information in the NHIRD was deidentified for patient privacy protection. This study adhered to the principles of the Declaration of Helsinki and the Reporting of Observational Studies in Epidemiology (STROBE) guidelines. Informed consent was waived by the Research Ethics Committee of the National Health Research Institute (EC1100703-E).

### 2.2. Study Population

All NHI enrollees aged 30 years or older, as of 1 January 2001, were identified for potential inclusion in this study (*n* = 12,540,002). Of these individuals, those who had an NHI record of exposure to CMs (*n* = 860,354), dementia (ICD-9-CM 290.0–290.4, 294.1, 331.0, or 331.1–331.2, *n* = 27,805), or stroke (ICD-9-CM 430–438, *n* = 271,025) and those who withdrew from the NHI program (*n* = 48,202) before 2001 were excluded. The remaining 11,332,616 NHI enrollees in 2001 were selected and divided into two groups: (1) those exposed to CMs (*n* = 3,993,770) and (2) those never exposed to CMs (*n* = 7,338,846) during the observation period (from 1 January 2001 to 31 December 2017). An iterative procedure was used to determine the status of exposure to CMs in each one-month time window to avoid immortal time bias. First, individuals who were exposed to CMs in January 2001 were identified as the exposed group; next, those remaining comprised the unexposed group and were selected through 1:1 matching by age, sex, comorbidities, and frequency of outpatient visits during the previous year. The date of exposure to CMs was defined as the index date for the exposed group, and the same index date was assigned to the corresponding matched subjects in the unexposed group. For the second month in February 2001, participants who had been selected as study pairs in the previous procedure and those who died in January 2001 were excluded, and the same procedure adopted in the first-month window was performed to identify matched pairs, and their index dates were assigned. The same procedure was repeated and continued in each one-month time window until the last observation month (December 2016).

### 2.3. Definitions of Outcome, Exposure, and Covariates

The primary outcome was the development of dementia, defined as ICD-9-CM 290.0–290.4, 294.1, 331.0, and 331.2 diagnoses in outpatient visits or during hospitalizations. The development of AD or VD was the secondary outcome. Patients diagnosed with ICD-9-CM 290.1 or 331.0 during outpatient department (OPD) visits or hospitalizations who did not have a history of stroke within the 3 years prior to the first diagnosis of AD were defined as having AD. Patients diagnosed with ICD-9-CM 290.4 during OPD visits or hospitalizations were defined as having VD. To avoid indication bias, the CM-exposed group comprised patients who underwent cardiac angiography or CM-enhanced computed tomography scanning other than during brain imaging examinations.

The comorbidities selected to balance the two investigated groups were hypertension (ICD-9-CM 401–405), hyperlipidemia (ICD-9-CM 272), diabetes (ICD-9-CM 250), chronic obstructive pulmonary disease (COPD, ICD-9-CM 490, 492, or 496), coronary artery disease (CAD, ICD-9-CM 410, 414, or 429.2), head injury (ICD-9-CM 959.01, 800–804, or 850–854), depression (ICD-9-CM 296.2, 296.3, 298.0, 300.4, 309.0, 309.1, 293.83, 296.90, 309.28, 296.82, or 311), and cancer (ICD-9-CM 140–208).

### 2.4. Statistical Analysis

The Kaplan–Meier method was used to compare the probability of being free of dementia over time between the CM-exposed and unexposed groups. The crude and multivariable-adjusted Cox proportional hazard models with robust sandwich standard error estimates were applied to compare the risk of developing dementia between the CM-exposed and unexposed groups. Survival analyses were conducted using an as-treated study design. The unexposed individuals were censored (i.e., their follow-up was stopped) if they received the CM after the index date. As the competing risks of incident dementia and death could have confounded the estimates of risk for our outcomes, we applied cause-specific hazard functions for adjustment. The results are presented as hazard ratios (HRs) with 95% confidence intervals (CIs) indicating the risk of exposure to CMs. The proportional hazard assumption was checked using the Schoenfeld residual test and log–log survival curves for all time-independent covariates. To calculate the risk of dementia, patients were censored at the time of death or at the end of the study, whichever came first. To reassure the robustness of our study results, we conducted a series of subgroup analyses by categorizing our study subjects into different subgroups according to their age, sex, and comorbidities.

A two-tailed *p*-value < 0.05 was considered statistically significant. SAS version 9.4 (SAS Institute Inc., Cary, NC, USA) and Stata SE version 11.0 (StataCorp, College Station, TX, USA) were used for the analyses.

## 3. Results

In total, 1,461,684 CM-exposed patients and 1,461,684 unexposed controls were enrolled in this retrospective study (Figure 1). Of the patients in the exposed group, 64.9% were men, 59.3% were aged ≤49 years, 10.3% had diabetes mellitus, 21.4% had hypertension, and 8.9% had hyperlipidemia (Table 1). The mean age of the patients in the exposure and control groups was 48.8 ± 13.3 years and 48.7 ± 13.3 years, respectively. The mean frequency of medical visits in the exposure and control groups was 14.4 (SD = 13.37) times per year. The prevalence of comorbidities associated with dementia, including diabetes, stroke, hypertension, hyperlipidemia, COPD, head injury, depression, and CAD, was comparable between the groups. In the Cox proportional hazard model with multivariable analysis after adjustment for age, sex, and comorbidities, patients exposed to CMs had a 2.09-fold higher risk of dementia compared with the controls (adjusted HR (aHR) = 2.09; 95% CI = 2.06–2.13). We observed that the relative risk of dementia was slightly lower in men (aHR = 0.84; 95% CI = 0.83–0.85) and higher in older patients (aHR = 1.12; 95% CI = 1.12–1.12). Other comorbidities were associated with a significantly higher risk of dementia, including diabetes (aHR = 1.35; 95% CI = 1.32–1.37), head injury (aHR = 1.33; 95% CI = 1.18–1.49), depression (aHR = 1.86; 95% CI = 1.77–1.95), and hypertension (aHR = 1.13; 95% CI = 1.12–1.15) (Table 2).

The risks for different types of dementia, including AD and VD, were calculated as aHRs through stratification according to the use of percutaneous coronary intervention (PCI) (Table 3). The results of the analysis indicated that patients without PCI who had been exposed to CMs had an incidence rate and aHR for AD of 1.55 per 1000 person-years and 1.84 (95% CI = 1.78–1.89), respectively, after adjustment for age, sex, and comorbidities. The same population had an incidence rate and aHR for VD of 1.02 per 1000 person-years and 4.70 (95% CI = 4.47–4.94), respectively. In patients who underwent PCI, the incidence rate and aHR for AD were 2.29 per 1000 person-years and 1.54 (95% CI = 1.36–1.76), respectively. Compared with the controls, the relative risk and aHR for VD were 1.60 per 1000 person-years and 3.96 (95% CI = 3.19–4.91), respectively, in the CM-exposed group. The results of the Kaplan–Meier analysis revealed a significantly lower disease-free rate of dementia in patients after exposure to CMs, compared with those not exposed to CMs (*p* < 0.001; Figure 2). The results of multivariable-stratified analyses indicated that the risk of AD was significantly higher in younger patients without underlying comorbidities; in this population, the risk was 3.50-fold higher in patients aged 30–49 years, 2.54-fold higher in patients without depression, 2.55-fold higher in patients without head injury, 2.57-fold higher in patients without diabetes mellitus, and 2.64-fold higher in patients without hypertension (Figure 3). The risk of VD was 6.92-fold higher in patients aged 30–49 years, 4.64-fold higher in those without depression, 4.65-fold higher in those without head injury, 4.73-fold higher in those without diabetes mellitus, and 5.09-fold higher in those without hypertension (Figure 4).

## 4. Discussion

CMs are extremely important in modern diagnostic medicine; however, concerns have been raised regarding their adverse effects, such as allergic reactions, hyperosmolarity, high viscosity, and organ injury, among which the CM-associated kidney injury is one of the most well-known and is caused by the systemic infusion of CMs.

A growing body of evidence suggests that SVD is significantly associated with various diseases, including those of the kidney and brain [11,12]. In both of these organs, the glomerular afferent arterioles of juxtamedullary nephrons or the cerebral perforating arteries are small, short vessels directly originating from large arteries [13,14]. Due to the similarities in vascular supply among peripheral end organs, which have low vascular resistance but receive large volumes of blood throughout cardiac cycles, the brain and the kidney are presumed to have high susceptibility to vascular injury owing to the similar anatomical and hemodynamic features of small artery diseases [14,15]. Additionally, both of these organs are vulnerable to some common risk factors, such as diabetes, hypertension, and hyperlipidemia, eventually resulting in a high prevalence of SVD, silent brain infarct, white matter pathology, and microbleeds [16]. Therefore, SVD in one organ can also occur in the other. In the present study, in which the effect of CMs on the brain was investigated, cofounders relevant to dementia were adjusted, and patients with a history of stroke and previous exposure to CMs were excluded. The results of the multivariable analysis revealed that patients exposed to CMs had a 2.09-fold increased risk of developing dementia after adjustment for age, sex, and presence of diabetes, hypertension, hyperlipidemia, head trauma history, CAD, cancer, and depression. In addition, diabetes and depression, generally regarded as risk factors for dementia, were associated with a higher risk of dementia.

Furthermore, the risks for specific types of AD and VD were separately determined. A significantly increased risk of VD was observed in patients after stratification according to whether they underwent PCI. Patients who underwent PCI after being exposed to CMs had a 4.70-fold increased risk of VD, which was statistically significant. Patients who did not undergo PCI, and who were generally regarded as having a lower cardiovascular risk, had a 3.96-fold increased risk of VD. The results of the analysis after stratification according to age revealed that younger patients aged 30–49 years had a remarkably increased risk of VD and AD after exposure to CMs. The trend was similar across all age groups and in other subgroups (Figure 3 and Figure 4). Similarly, the risk of dementia was still significantly higher in patients without any underlying disease or comorbidity.

Taken together, our findings suggested that CM exposure is significantly associated with the risk of dementia and independently associated with traditional cardiovascular factors such as hypertension, diabetes, hyperlipidemia, and smoking.

Recent biological evidence suggests that exposure to CMs is likely to trigger an inflammatory response, reporting that canonical Nlrp3 inflammasome was activated in macrophages after the infusion of a contrast agent in a mouse model. Moreover, levels of inflammasome biomarker IL-18 and caspase have been reported to increase briefly in patients after their subjection to coronary angiography [17,18]. Another study suggested that Nlrp3 inflammasome in oxidative-stress-induced neuroinflammation is associated with AD in that it promotes the maturation of proIL-1 and proIL-18 and that microglial cells are the resident macrophages of the central nervous system (CNS) and are primarily responsible for the immune system of the CNS [19]. Several neurodegenerative disorders, including AD, Parkinsonism, and amyotrophic lateral sclerosis, involve the immune system, primarily microglial cells. These cells adopt a signature of chronic proinflammatory cytokines, which can cause the degeneration of neural cells, referred to as demyelination, which is a common feature of neurodegenerative disorders such as AD and multiple sclerosis [19,20]. Moreover, CMs are associated with activation of the high-mobility group box 1 pathway and NF-kB transcriptional factors, which are crucial for various inflammatory cytokines such as TNF-a, IL-1, and IL-6 [21].

Considerable evidence suggests the involvement of inflammasome-mediated inflammatory pathways in CNS disorders and the associated behavioral changes. Neuroinflammation, which is an innate immune response to harmful and irritable insults such as pathogens and metabolic toxic waste, is mediated by protein complexes known as inflammasomes; these inflammasomes play instrumental roles in neuroinflammation-related consequences, including neurodegenerative diseases, cognitive impairment, and dementia [22]. In particular, the NLRP inflammasome plays a role in neurodegenerative disorders such as AD [23].

The characteristics of the pathology and behavioral deficits of chronic neuroinflammation do not manifest until an advanced age. This has been attributed to the capacity of the brain to compensate for the presence of chronic neuroinflammation by regulating the glutamatergic system [24]. This suggests that neuroinflammation in itself does not cause AD; however, it acts as an initiator, enhancer, and sustainer of AD during old age, which is reinforced by various other etiologies [22]. Our findings suggested that the association of CM exposure with the risk of dementia may be considered an example of inflammasome-related SVD of the brain.

Although our results are robust, this study has some limitations. First, detailed personal data on tobacco smoking status, alcohol consumption, and family history of dementia were unavailable in the database. To account for these potential confounders, which can influence the risk of dementia, proxy variables such as the incidence of COPD in smokers, hypertension, hyperlipidemia, and diabetes were included. Second, patients exposed to CMs were identified by the procedure codes associated with the CMs; therefore, the dosages of the CMs to which the patients were exposed could not be estimated in this study and may have been underestimated for patients receiving treatment for some conditions that may involve exposure to larger dosages of CMs, such as intervention therapy with angiography. Third, clinical information regarding dementia, including disease severity, mental health status, disease duration, predominant syndromes, and neurological image data, could not be accessed in the NHIRD. However, dementia was diagnosed by neurology specialists on the basis of the standard diagnostic criteria for the disease. Fourth, with regard to kidney injury, kidney function could be easily measured during laboratory examinations according to factors such as blood urea level, creatinine level, or urine volume. However, clinical cognitive impairment manifesting as brain changes could not be easily recognized, and the risk of dementia could have been underestimated in the present study. Although we controlled for all known confounding factors in our study, some unknown inherent confounders might have biased the results because of the study’s retrospective case–control design, which generally involves fewer data than a randomized control design.

## 5. Conclusions

To the best of our knowledge, this study is the first to report that exposure to CMs is significantly associated with the risk of developing dementia. In addition to the detrimental effect of CM exposure on the kidney, CM-associated brain injuries such as dementia should receive more attention. Further studies on the effect of CMs on the brain are warranted.

## Figures and Tables

**Figure 1 biomedicines-10-02015-f001:**
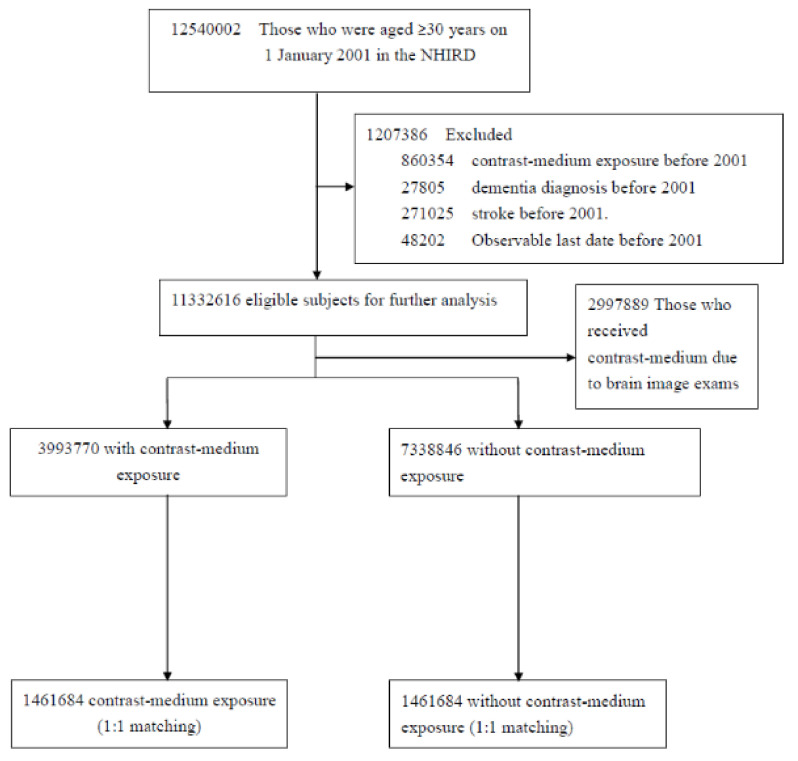
Flowchart of patient selection for the study.

**Figure 2 biomedicines-10-02015-f002:**
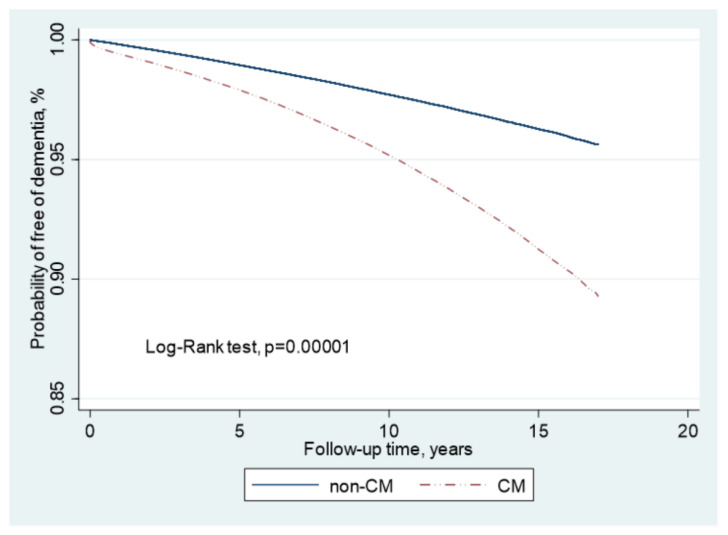
Results of Kaplan–Meier analysis of dementia-free rate in patients with and without exposure to contrast medium.

**Figure 3 biomedicines-10-02015-f003:**
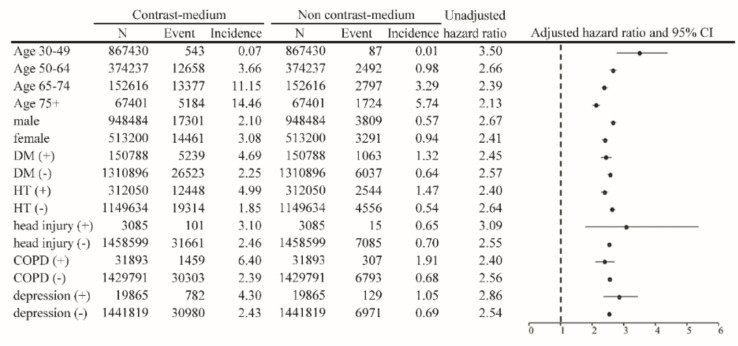
Results of multivariable-stratified analyses on the risk of Alzheimer’s disease among patients with and without exposure to contrast medium.

**Figure 4 biomedicines-10-02015-f004:**
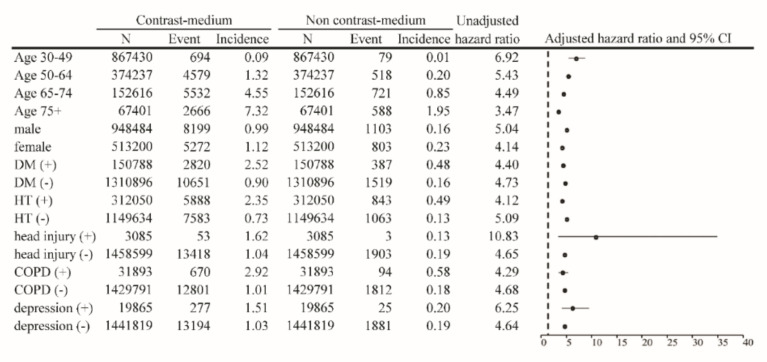
Results of multivariable-stratified analyses on the risk of vascular dementia among patients with and without exposure to contrast medium.

**Table 1 biomedicines-10-02015-t001:** Baseline characteristics of participants with and without exposure to contrast medium.

	Before Match					After Match				
	Contrast Medium Exposure		Contrast Medium Non-Exposure		*Standard Difference*	Contrast Medium Exposure		Contrast Medium Non-Exposure		*Standard Difference*
	**N (3,993,770)**	**%**	**N (7,338,846)**	**%**		**N (1,461,684)**	**%**	**N (1,461,684)**	**%**	
Age										
Mean (SD)	51.8	13.8	45.5	12.8	0.47957	48.8	13.3	48.7	13.3	0.00309
30–49	1,970,025	49.3	5,208,013	70.9	−0.45321	867,430	59.3	867,430	59.3	0.0000
50–64	1,170,274	29.3	1,400,847	19.1	0.24012	374,237	25.6	374,237	25.6	0.0000
65+	853,471	21.4	729,986	10.0	0.31819	220,017	15.1	220,017	15.1	0.0000
Sex										
Male	2,264,465	56.7	3,500,813	47.7	0.18091	948,484	64.9	948,484	64.9	0.0000
Female	1,729,305	43.3	3,838,033	52.3	−0.18091	513,200	35.1	513,200	35.1	0.0000
Frequency of outpatient visits/per year, mean(SD)	13.8	14.9	9.6	11.7	0.31432	14.4	13.7	14.4	13.7	0.0000
Comorbidity										
Hypertension	642,508	16.1	521,842	7.1	0.28323	312,050	21.4	312,050	21.4	0.0000
Hyperlipidemia	212,278	5.3	172,815	2.4	0.15512	130,009	8.9	130,009	8.9	0.0000
Diabetes	290,025	7.3	216,803	3.0	0.19677	150,788	10.3	150,788	10.3	0.0000
COPD	94,784	2.4	81,458	1.1	0.09647	31,893	2.2	31,893	2.2	0.0000
CAD	165,821	4.2	114,257	1.6	0.15599	8,0411	5.5	80,411	5.5	0.0000
Head injury	22,432	0.6	29,521	0.4	0.02315	3085	0.2	3085	0.2	0.0000
Depression	56,470	1.4	65,324	0.9	0.04879	19,865	1.4	19,865	1.4	0.0000
Cancer	54,699	1.4	30,911	0.4	0.10100	33,023	2.3	33,023	2.3	0.0000

COPD: chronic obstructive pulmonary disease; CAD: coronary artery disease.

**Table 2 biomedicines-10-02015-t002:** Risk of development of dementia in association with exposure to contrast medium, sex, age, and comorbidities in univariable and multivariable Cox regression models.

	Dementia						
	Incidence	Crude		Adjusted ^†^		Adjusted ¶	
Variable	Rate	HR	(95% CI)	HR	(95% CI)	HR	(95% CI)
Exposure of contrast medium	5.49	2.25	(2.21–2.28) ***	2.09	(2.06–2.13) ***	2.09	(2.06–2.13) ***
Sex (Men vs. Women)	3.58	0.72	(0.71–0.73) ***	0.84	(0.83–0.85) ***	0.84	(0.83–0.85) ***
Age, years							
50–64	5.01	19.36	(18.71–20.04) ***	18.00	(17.39–18.62) ***	18.00	(17.39–18.63) ***
65–74	19.03	77.02	(74.46–79.66) ***	69.10	(66.78–71.50) ***	69.10	(66.77–71.51) ***
75+	36.94	161.81	(156.27–167.55) ***	149.92	(144.72–155.31) ***	149.92	(144.66–155.37) ***
Baseline comorbidities (yes vs. no)							
Hypertension	8.93	3.11	(3.07–3.15) ***	1.13	(1.12–1.18) ***	1.13	(1.12–1.15) ***
Hyperlipidemia	5.85	1.55	(1.52–1.59) ***	0.97	(0.95–0.99) **	0.97	(0.95–0.99) **
Diabetes	8.63	2.48	(2.44–2.52) ***	1.35	(1.32–1.37) ***	1.35	(1.32–1.37) ***
COPD	12.93	3.37	(3.27–3.47) ***	1.13	(1.10–1.16) ***	1.13	(1.10–1.17) ***
CAD	10.72	2.91	(2.86–2.97) ***	0.99	(0.97–1.01)	0.99	(0.97–1.01)
Head injury	5.83	1.35	(1.21–1.51) ***	1.33	(1.19–1.48) ***	1.33	(1.18–1.49) ***
Depression	6.93	1.72	(1.65–1.80) ***	1.86	(1.78–1.94) ***	1.86	(1.77–1.95) ***
Cancer	6.16	1.51	(1.45–1.57) ***	0.82	(0.79–0.85) ***	0.82	(0.79–0.85) ***

Incidence rate (per 1000 person-years): exposed group. Crude HR: crude hazard ratio. ^†^ Multivariable models included all statistically significant risk factors in the univariable Cox model. ¶ Competing risk and robust sandwich standard error estimates. ** *p* < 0.01, *** *p* < 0.001.

**Table 3 biomedicines-10-02015-t003:** Incidence rates and hazard ratios of Alzheimer’s disease and vascular dementia due to exposure to contrast medium.

		Alzheimer’s Disease			Vascular Dementia			
	Incidence Rate	Crude HR	Adjusted HR †	Adjusted HR ¶	Incidence Rate	Crude HR	Adjusted HR †	Adjusted HR ¶
Contrast medium exposure (overall)	1.57	2.10(2.04–2.16) ***	1.82(1.77–1.88) ***	1.82(1.77–1.88) ***	1.04	5.18(4.93–5.43) ***	4.66(4.44–4.89) ***	4.66(4.44–4.89) ***
PCI-unrelated contrast medium exposure	1.55	2.11(2.05–2.17) ***	1.84(1.78–1.89) ***	1.84(1.78–1.89) ***	1.02	5.20(4.95–5.46) ***	4.70(4.47–4.94) ***	4.70(4.47–4.94) ***
PCI-related contrast medium exposure	2.29	1.75(1.54–2.00) ***	1.54(1.35–1.76) ***	1.54(1.36–1.76) ***	1.60	4.42(3.56–5.48) ***	3.96(3.19–4.91) ***	3.96(3.19–4.91) ***

Incidence rate (per 1000 person-years): exposed group. † Multivariable models included all statistically significant risk factors in the univariable Cox model. ¶ Competing risk and robust sandwich standard error estimates. *** *p* < 0.001.

## Data Availability

The dataset(s) supporting the conclusions of this article is managed by Taiwan’s Ministry of Health and Welfare (MOHW). The MOHW approved our application to access these data. Any investigator interested in accessing this dataset must submit an application to the MOHW. The MOHW address is No. 488, Sec. 6, Zhongxiao E. Rd., Nangang Dist., Taipei City 115, Taiwan (R.O.C.). Phone: +886-2-8590-6848. Please contact MOHW personnel (email: stcarolwu@mohw.gov.tw) for further assistance. All the relevant data are provided in this paper.

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
