# Peer review of "Risk of Dementia after Exposure to Contrast Media: A Nationwide, Population-Based Cohort Study"

_biomedicines, 2022, doi:10.3390/biomedicines10082015_

Round 1

Reviewer 1 Report

In this paper, the authors investigated the adverse effect of contrast medium on brain thanks to a very large cohort. In total, 11,332,616 NHI enrollees were selected and divided into two groups, exposed and not exposed to contrast medium during the observation period. Totally, 1,461,684 patients exposed to contrast medium and 1,461,684 controls not exposed were enrolled. Using multivariable Cox proportional hazards models, the authors showed that patients exposed to contrast medium had an overall 2.09-fold higher risk of dementia, and using multivariable stratified analyses, that the risk of Alzheimer’s disease was remarkably high in younger patients without any underlying comorbidity. Moreover, the authors observed a 4-fold increase risk of vascular dementia after exposure to contrast medium.

The study is very impressive thanks to a very large cohort collected with the National Health Insurance (NHI) program. The selection process of 1,461,684 pairs amongst 11,332,616 NHI enrollees represents a huge amount of work. The topic is very interesting and I agree with the authors: “Further studies on the influence of exposure to contrast medium on the brain are warranted”.

The paper is well written and structured.

The statistical methods are very sophisticated and therefore difficult to control or criticize. However the authors have clearly indicated the statistical pipeline.

My remarks are the following:

Remark 1. Limitations

The authors should add a section “Limitation”, which is currently at the end of the section “Discussion”.

Remark 2. ODP acronym

In section “2.3. Definitions of outcome, exposure, and covariates”, the ODP acronym was used without be spelled out.

Remark 3. “2.2. Study population”

The following sentence was unclear: “All NHI enrollees aged 30 years or older as of January 1, 2001, were identified for potential inclusion in this study (n = 12,540,002). Of these individuals, those who had an NHI record of exposure to CM (n = 860,354), dementia (ICD-9-CM 290.0–290.4, 294.1, 331.0, or 331.1–331.2, n = 27,805), or stroke (ICD-9-CM 430–438, n = 271,025) and those who withdrew from the NHI program (n = 48,202) before 2001 were excluded”.

It is unclear why those of the 12,540,002 NHI enrollees, who had an NHI record of exposure to CM (n = 860,354) were excluded.

Author Response

Remark 1. Limitations

The authors should add a section “Limitation”, which is currently at the end of the section “Discussion”.

Answer: Thank you so much for the suggestion. The Limitation has been inserted in Page 12 as below:

“Although our results are robust, this study has some limitations. First, detailed personal information on tobacco smoking status, alcohol consumption, and family history of dementia was unavailable in the database. To account for these potential confounders, which can influence the risk of dementia, proxy variables such as the incidence of COPD in smokers, hypertension, hyperlipidemia, and diabetes were included. Second, patients exposed to CM were identified by the procedure codes associated with CM; therefore, the dosage of CM that the patients were exposed to could not be estimated in this study and may have been underestimated for patients receiving treatment for some conditions that may involve exposure to larger dosages of CM, such as intervention therapy with angiography. Third, clinical information regarding dementia, including disease severity, mental health status, disease duration, predominant syndromes, and neurological image data, could not be accessed in the NHIRD. However, dementia was diagnosed by neurology specialists on the basis of the standard diagnostic criteria for the disease. Fourth, with regard to kidney injury, kidney function could be easily measured during laboratory examinations according to factors such as blood urea level, creatinine level, or urine volume. However, clinical cognitive impairment manifesting as brain changes could not be easily recognized, and the risk of dementia could have been underestimated in the present study. Although we controlled for all known confounding factors in our study, some unknown inherent confounders might have biased the results because of the study’s retrospective case–control design, which generally involves less data than does a randomized control design.”

Remark 2. ODP acronym

In section “2.3. Definitions of outcome, exposure, and covariates”, the ODP acronym was used without be spelled out.

 Answer:  Thank you for the comment.

 It has been revised as outpatient department (OPD)

Remark 3. “2.2. Study population”

The following sentence was unclear: “All NHI enrollees aged 30 years or older as of January 1, 2001, were identified for potential inclusion in this study (n = 12,540,002). Of these individuals, those who had an NHI record of exposure to CM (n = 860,354), dementia (ICD-9-CM 290.0–290.4, 294.1, 331.0, or 331.1–331.2, n = 27,805), or stroke (ICD-9-CM 430–438, n = 271,025) and those who withdrew from the NHI program (n = 48,202) before 2001 were excluded”.

It is unclear why those of the 12,540,002 NHI enrollees, who had an NHI record of exposure to CM (n = 860,354) were excluded.

Answer: Thank you for the comment.  Before 2001, DATA from NHI was insufficiency and the insurance cover rate was relatively low. Therefore, the study period was set up at the year of 2001 which cover rate approximately 99% in Taiwan.

Reviewer 2 Report

First of all,I would like to thank the authors. The topic they have selected is very interesting, because we have to continue learning about potential risk factors for development of Alzheimer's disease dementia. 

Now, I am going to notify my concerns related to this study: 

1) Small vessel disease is indeed related to AD dementia, indecently to classic cardiovascular risk factors. Have you checked of AD dementia risk in relation to Fazekas scale across both groups? 

2) Where differences in neuroimaging (hippocampal volume, parietal atrophy, leucoraiosis, microhemorrages...) between groups?

All of them could be indicating AD presymtomatic stage.

3) Diagnosis of AD was only baed in clinical criteria? Or was based In biomarkers at least in some cases? Could yo extend this point. 

4) Risk is related to dementia AD but not to Mild Cognitive Impairment. AD is a continuum from a very large presymptomatic stage to a prodromal stage years before to dementia stage. 

5) Cogntion and risk of AD could be influenced also by APOE status, education level, epilepsy condition (associated with amyloid deposition)... Extend the discussion at this point.

6) Try to elaborate the relation specifically with AD core anatomophatological hallmarks. Inflammation is relevant in AD but there are lot more mechanisms. 

Try to add a figure to explain the possible mechanisms that could be associated. 

7) Be careful with the categorical statements. You are suggesting a potential link (not so evident and clear) for various diseases (not diagnosed with gold-standard technique), all of them with a lot of risk factors associated.

I am not going to continue with the review until I know the answer of the authors to this comments. 

Author Response

1) Small vessel disease is indeed related to AD dementia, indecently to classic cardiovascular risk factors. Have you checked of AD dementia risk in relation to Fazekas scale across both groups? 

Answer: Thank you for the comment.

Clinical information regarding dementia, including disease severity, mental health status, disease duration, predominant syndromes, and neurological image data, could not be accessed in the NHIRD. However, dementia was diagnosed by neurology specialists on the basis of the standard diagnostic criteria for the disease.

2) Where differences in neuroimaging (hippocampal volume, parietal atrophy, leucoraiosis, microhemorrages...) between groups?

All of them could be indicating AD presymtomatic stage.

Answer: Thank you for the comment.

Clinical information regarding dementia, including disease severity, mental health status, disease duration, predominant syndromes, and neurological image data, could not be accessed in the NHIRD. However, dementia was diagnosed by neurology specialists on the basis of the standard diagnostic criteria for the disease.

3) Diagnosis of AD was only baed in clinical criteria? Or was based In biomarkers at least in some cases? Could yo extend this point. 

Answer: Thank you for the comment.

The diagnosis of dementia was diagnosed by neurology specialists on the basis of the standard diagnostic criteria for the disease.

4) Risk is related to dementia AD but not to Mild Cognitive Impairment. AD is a continuum from

a very large presymptomatic stage to a prodromal stage years before to dementia stage. 

Answer: Thank you for the comment.

In the present study, the diagnosis of dementia was diagnosed by neurology specialists on the basis of the standard diagnostic criteria for the disease.However, clinical cognitive impairment manifesting as brain changes could not be easily recognized, and the risk of dementia could have been underestimated in the present study.

5) Cognition and risk of AD could be influenced also by APOE status, education level, epilepsy condition (associated with amyloid deposition)... Extend the discussion at this point.

Answer: Thank you for the comment.

In the present study, clinical information regarding dementia, including disease severity, mental health status, disease duration, predominant syndromes, and neurological image data, could not be accessed in the NHIRD. However, clinical cognitive impairment manifesting as brain changes could not be easily recognized, and the risk of dementia could have been underestimated in the present study. Although we controlled for all known confounding factors in our study, some unknown inherent confounders might have biased the results because of the study’s retrospective case–control design, which generally involves less data than does a randomized control design.

6) Try to elaborate the relation specifically with AD core anatomophatological hallmarks. Inflammation is relevant in AD but there are lot more mechanisms. 

Try to add a figure to explain the possible mechanisms that could be associated. 

Answer: Thank you for the suggestion.

7) Be careful with the categorical statements. You are suggesting a potential link (not so evident and clear) for various diseases (not diagnosed with gold-standard technique), all of them with a lot of risk factors associated.

Answer: Thank you for the comment.
